# Perception of a conserved family of plant signalling peptides by the receptor kinase HSL3

Jack Rhodes[1], Andra-Octavia Roman[2], Marta Bjornson[3†], Benjamin Brandt[3‡], Paul Derbyshire[1], Michele Wyler[4], Marc W Schmid[4], Frank LH Menke[1], Julia Santiago[2], Cyril Zipfel[1,3]*

[1]The Sainsbury Laboratory, University of East Anglia, Norwich, United Kingdom; [2]The Plant Signaling Mechanisms Laboratory, Department of Plant Molecular Biology, University of Lausanne, Lausanne, Switzerland; [3]Institute of Plant Molecular Biology, Zurich-Basel Plant Science Center, University of Zurich, Zurich, Switzerland; [4]MWSchmid GmbH, Glarus, Switzerland

**\*For correspondence:**
cyril.zipfel@botinst.uzh.ch

**Present address:** [†]Department of Plant Sciences, University of California Davis, Davis, United States; [‡]Biozentrum der Ludwig-Maximilians-Universität München, Department Biologie I – Botanik, Planegg-Martinsried, Germany

**Competing interest:** The authors declare that no competing interests exist.

**Abstract** Plant genomes encode hundreds of secreted peptides; however, relatively few have been characterised. We report here an uncharacterised, stress-induced family of plant signalling peptides, which we call CTNIPs. Based on the role of the common co-receptor BRASSINOSTEROID INSENSITIVE 1-ASSOCIATED KINASE 1 (BAK1) in CTNIP-induced responses, we identified in *Arabidopsis thaliana* the orphan receptor kinase HAESA-LIKE 3 (HSL3) as the CTNIP receptor via a proteomics approach. CTNIP-binding, ligand-triggered complex formation with BAK1, and induced downstream responses all involve HSL3. Notably, the HSL3-CTNIP signalling module is evolutionarily conserved amongst most extant angiosperms. The identification of this novel signalling module will further shed light on the diverse functions played by plant signalling peptides and will provide insights into receptor-ligand co-evolution.

## Editor's evaluation

Beginning with transcriptome data, Rhodes et al., identify a new family of peptides with signalling function called CTNIP in the model plant *Arabidopsis thaliana*. They use an elegant biochemical capture approach to pinpoint an LRR receptor kinase called HSL3 as the only receptor for these peptides. They provide convincing genetic and biochemical evidence that HSL3 binds CTNIP and that CTNIP perception triggers HSL3-dependent cytoplasmic calcium influx, ROS production, and transcriptional changes. Furthermore, they provide initial evidence that the CTNIP-HSL3 module may participate in regulating root growth.

## Introduction

Secreted plant signalling peptides play major roles in growth, development, and stress responses (*Olsson et al., 2019*). Whilst many hundreds of signalling peptides are predicted to be encoded in plant genomes, relatively few have been characterised and their corresponding receptors are mostly unknown (*Boschiero et al., 2020*; *Ghorbani et al., 2015*; *Olsson et al., 2019*).

Most signalling peptides are recognised by cell-surface localised receptors, especially by leucine-rich repeat receptor kinases (LRR-RKs). LRR-RKs generally function through the ligand-dependent recruitment of a shape complementary co-receptor to form an active signalling complex (*Hohmann et al., 2017*). The best characterised peptide receptors belong to LRR-RK subfamily XI, these receptors

recognise distinct families of plant peptides involved in growth, development, or stress responses (*Furumizu et al., 2021*). Notably, the LRR-RK MIK2, which belongs to the closely related LRR-RK subfamily XIIb (an outgroup recently included within subfamily XI; *Liu et al., 2017*; *Man et al., 2020*) was recently shown to perceive SCOOP peptides (*Hou et al., 2021*; *Rhodes et al., 2021*). Despite intensive studies on the LRR-RK subfamily XI, the ligand for HAESA-like 3 (HSL3) has remained elusive, hindering our ability to investigate peptide-receptor co-evolution across the family (*Furumizu et al., 2021*; *Liu et al., 2020*).

## Results and discussion

Several peptides (PEPs, PIPs, SCOOPs, CLEs, and IDLs) recognised by LRR-RKs from subfamily XI or XIIb are transcriptionally up-regulated by abiotic or biotic stresses (*Bartels et al., 2013*; *Gully et al., 2019*; *Kim et al., 2021*; *Takahashi et al., 2018*; *Vie et al., 2015*). In order to identify novel stress-induced signalling peptides, we searched for *Arabidopsis thaliana* (hereafter, *Arabidopsis*) transcripts encoding short proteins (<150 amino acids) with a predicted signal peptide, which were induced upon biotic elicitor treatment (*Supplementary file 1*; *Bjornson et al., 2021*). Through this analysis, we identified an uncharacterised family of peptides with five predicted members, which we named CTNIP1–5 (pronounced catnip; AT1G06135, AT1G06137, AT2G31335, AT2G31345, and AT3G23123, respectively) based on relatively conserved residues within the peptides (*Figure 1a–b*; *Figure 1—figure supplement 1a, b*).

To determine whether CTNIPs function as signalling peptides, we synthetised peptides corresponding to the whole CTNIP proteins excluding the predicted signal peptide. CTNIP1–4 peptides were able to induce cytoplasmic $Ca^{2+}$ influx and mitogen-activated protein kinase (MAPK) phosphorylation – hallmarks of peptide signalling (*Figure 1c–e*). However, a synthetic peptide derived from the divergent CTNIP5 peptide was inactive (*Figure 1—figure supplements 1a and 2c*). Notably, the C-terminal 23 amino acids of CTNIP4 (CTNIP4[48-70]) were sufficient to induce responses (*Figure 1—figure supplement 2a, b*), suggesting that the minimal bioactive peptide is contained within this region. Notably, this region contains two highly conserved cysteine residues (*Figure 1b*). Mutation of these cysteine residues revealed they are required for CTNIP4 activity (*Figure 1—figure supplement 2c*). The exact sequence of the mature peptides produced in planta, as well as their secretion and cleavage mechanisms however require future validation. Going forward, we focused on CTNIP4 as a representative member of this peptide family, as its transcript was the most up-regulated upon elicitor treatment (*Figure 1a*).

We hypothesised that CTNIPs may be perceived by a cell-surface LRR-receptor. Typically LRR-receptors are dependent upon the SOMATIC EMBRYOGENESIS RECEPTOR KINASE (SERK) family of co-receptors (*Hohmann et al., 2017*). We therefore tested whether CTNIP-induced responses were affected in *bak1-5*, an allele of *BRASSINOSTEROID INSENSITIVE 1-ASSOCIATED KINASE 1* (*BAK1/ SERK3*) that has a dominant-negative impact on SERK signalling (*Perraki et al., 2018*; *Schwessinger et al., 2011*). Concordant with perception by an LRR-receptor, we observed significantly impaired CTNIP4-induced reactive oxygen species (ROS) production in *bak1-5* (*Figure 1f–g*).

Ligand binding induces receptor-SERK heterodimerisation to activate signalling (*Hohmann et al., 2017*). To identify the CTNIP receptor, we therefore employed *Arabidopsis* lines expressing BAK1 tagged with green fluorescent protein (GFP) as a molecular bait to identify the CTNIP receptor. Using affinity purification followed by mass spectrometry (*Saur et al., 2016*), we looked for proteins specifically enriched into the BAK1 complex upon CTNIP4 treatment (*Figure 2a*). In four independent biological replicates, the protein most enriched in the BAK1 complex upon CTNIP4 treatment was the LRR-RK HSL3 (*Figure 2b*; *Figure 2—figure supplement 1*; *Supplementary file 2*), making this a promising candidate for being the CTNIP receptor. We could independently confirm CTNIP-induced HSL3-BAK1 complex formation by affinity purification (*Figure 2c*).

Consistent with a receptor function, the HSL3 ectodomain (HSL3[ECD], residues 22–627) heterologously expressed in insect cells could directly bind CTNIP4 with a dissociation constant of ~4 µM in in vitro binding assays using isothermal titration calorimetry (*Figure 2d–e*; *Figure 2—figure supplement 2a, b*). In the presence of CTNIP4, BAK1 bound HSL3 with a dissociation constant in the submicromolar range (0.392 µM) (*Figure 2d–f*; *Figure 2—figure supplement 2b*), consistent with its role as co-receptor. Furthermore, the two conserved cysteine residues are required for receptor binding

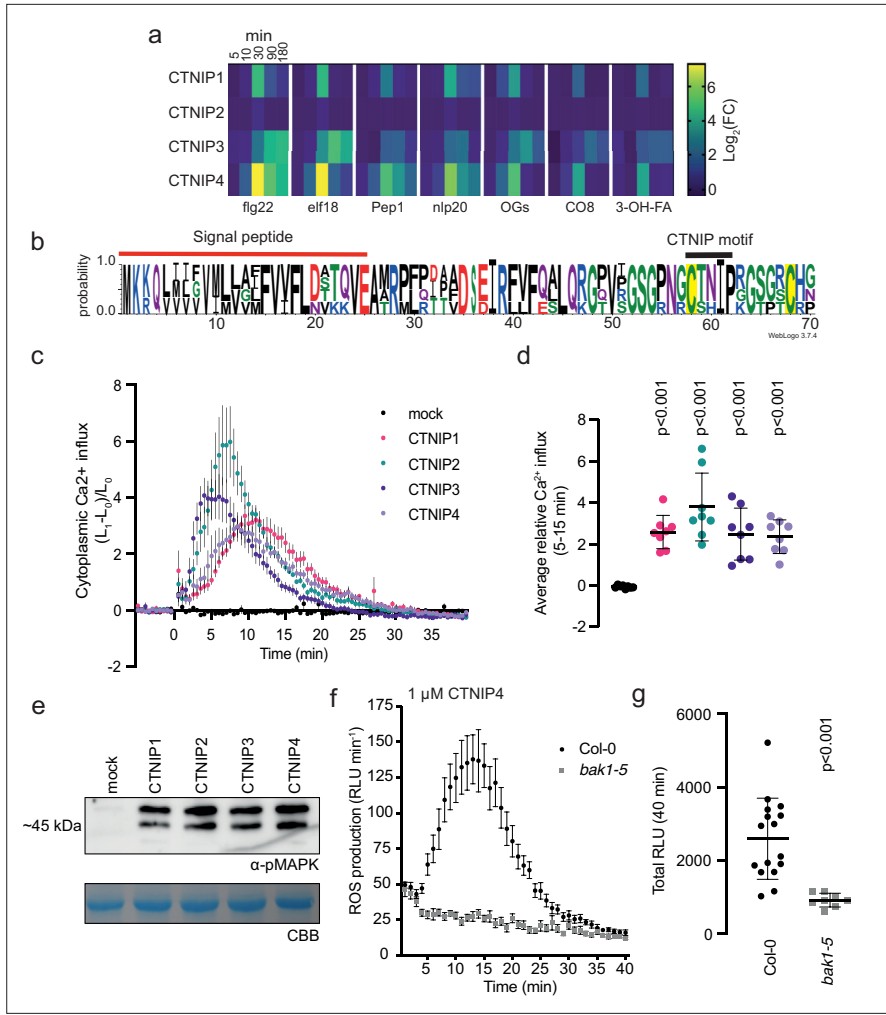

**Figure 1.** CTNIPs are a novel family of plant signalling peptide. (**a**) Heat map showing $\log_2$(FC) expression levels of CTNIP1–4 in response to a range of elicitors (data obtained from **Bjornson et al., 2021**). CTNIP5 was excluded as it is unannotated in the TAIR10 annotation, which was used to map the RNA sequencing reads. (**b**) Sequence probability logo from *Arabidopsis* CTNIP1–4 generated using WebLogo3. Signal peptide (as predicted by SignalP5.0) and CTNIP motif are indicated, and conserved cysteine residues are highlighted in yellow. Amino acids are coloured based on their biochemical properties: red = acidic; blue = basic; black = hydrophobic, and green = polar. (**c–d**) Cytoplasmic calcium influx measured in *p35S::AEQUORIN* seedlings after treatment with 1 µM CTNIP relative to pre-treated levels (n = 8 seedlings). (**c**) Points represent mean; error bars represent SEM. (**d**) Represents mean relative $Ca^{2+}$ influx between 5 and 15 min. A line represents mean; error bars represent SD. p-Values indicate significance relative to the wild-type (WT) control in a Dunnett's multiple comparison test following one-way ANOVA. (**e**) Western blot using α-p42/p44-ERK recognising phosphorylated MAP kinases in seedlings treated with 100 nM CTNIPs or mock for 15 min. The membrane was stained with CBB, as a loading control. (**f–g**) ROS production in leaf disks collected from 4-week-old *Arabidopsis* plants induced by 1 µM CTNIP4 application (n ≥ 8). (**f**) Points represent mean; error bars represent SEM. (**g**) Integrated ROS production over 40 min. Line represents mean; error bars represent SD. p-Values indicate significance relative to the WT control in a two-tailed t-test. All experiments were repeated and analysed three times with similar results. ROS, reactive oxygen species; CBB, Coomassie brilliant blue.

The online version of this article includes the following source data and figure supplement(s) for figure 1:

**Source data 1.** CTNIPs are a novel family of plant signalling peptide.

**Source data 2.** CTNIP-induced MAPK phosphorylation.

**Figure supplement 1.** Alignment and phylogeny of *Arabidopsis* CTNIPs.

**Figure supplement 1—source data 1.** Expression levels of *CTNIP1, CTNIP2, CTNIP3,* and *CTNIP5.*

**Figure supplement 2.** Characterisation of CTNIP synthetic peptides.

*Figure 1 continued*

**Figure supplement 2—source data 1.** CTNIP-induced calcium influx.

**Figure supplement 2—source data 2.** CTNIP-induced MAPK phosophorylation.

and co-receptor recruitment explaining their loss of signalling activity (*Figure 2d and g–h*; *Figure 1—figure supplement 2c*; *Figure 2—figure supplement 2b*).

Notably, we were unable to detect binding of INFLORESCENCE DEFICIENT IN ABSCISSION (IDA), the ligand for the related receptors HAESA and HSL2 (*Meng et al., 2016*; *Santiago et al., 2016*), to HSL3$^{ECD}$ (*Figure 2d*; *Figure 2—figure supplement 2b*), demonstrating distinct ligand specificity. Accordingly, structural analysis of an HSL3$^{ECD}$ homology model reveals that the HSL3 receptor lacks key conserved motifs required to recognise IDA peptides (*Figure 2—figure supplement 3*; *Roman et al., 2022*; *Santiago et al., 2016*). Together, our data show that, while HSL3 is phylogenetically related to HAE, HSL1, and HSL2, it perceives distinct peptides (i.e. CTNIPs) most likely via different binding interfaces, which remains to be investigated in future structural studies.

Having established biochemically that HSL3 is the CTNIP receptor, we tested its genetic requirement for CNTIP-induced responses. As expected, we found that HSL3 is strictly required for CTNIP-induced MAPK phosphorylation and whole genome transcriptional reprogramming (*Figure 3a–b*; *Figure 3—figure supplement 1*). Notably, whilst 30 min treatment with 100 nM CTNIP4 led to differential expression of 1074 genes in wild-type Col-0, none were differentially expressed in *hsl3-1* (p<0.05, |log$_2$(FC)|>1) (*Figure 3b*; *Supplementary file 3*).

We could additionally show that transient expression of HSL3 in *Nicotiana benthamiana* is sufficient to confer responsiveness to CTNIPs (*Figure 3c*). Furthermore, whilst 500 nM CTNIP4 was unable to significantly inhibit growth in Col-0 seedlings, plants that overexpress *HSL3* became hypersensitive to active CTNIP4 (*Figure 3d–e*; *Figure 3—figure supplement 2*). Taken together, our biochemical and genetic results demonstrate that HSL3 is the CTNIP receptor.

CTNIPs induce general early signalling outputs indicative of RK signalling, including cytoplasmic Ca$^{2+}$ influx, MAPK phosphorylation, and ROS production (*Figure 1*; *Olsson et al., 2019*). In addition, CTNIP4 treatment induces significant HSL3-dependent transcriptional reprogramming (*Figure 3b*). Consistent with the up-regulation of *CTNIP* and *HSL3* expression by biotic elicitors (*Figure 1a*; *Figure 2—figure supplement 1*), gene ontology analysis highlighted the enrichment of many defence- and stress-responsive pathways upon CTNIP4 treatment (*Supplementary file 4*). This is a pattern shared with other biotic elicitors (*Figure 3—figure supplement 3*) indicative of a general stress response (*Bjornson et al., 2021*).

To investigate the biological consequence of HSL3 signalling, we fused the extracellular and transmembrane domains of BAK1-INTERACTING RECEPTOR-LIKE KINASE 3 (BIR3) to the cytoplasmic domain of HSL3 under the control of the *HSL3* promoter (*Figure 3—figure supplement 4a*). This chimeric approach allows constitutive complex formation with SERKs, thus mimicking constitutive activation of an endogenous receptor kinase (*Hohmann et al., 2020*). Transgenic lines expressing this chimeric construct exhibited developmental defects, notably enhanced root curling (*Figure 3f*; *Figure 3—figure supplement 4a*). This phenotype is dependent upon SERK binding as the phenotype is abolished when residues essential for SERK binding are mutated (*Figure 3—figure supplement 4b*; *Hohmann et al., 2020*). In addition, CTNIP4 treatment inhibited root growth and induced root skewing in an HSL3-dependent manner (*Figure 3g–i*; *Figure 3—figure supplement 4c*). Furthermore, *CTNIP4* overexpression, either with or without a C-terminal tag, was sufficient to induce a similar phenotype (*Figure 3j–k*; *Figure 3—figure supplement 4d*). These data suggest that the HSL3-CTNIP signalling module modulates root development, similar to other signalling peptides recognised by LRR-RK subfamily XI members (*Jeon et al., 2021*; *Jourquin et al., 2020*). Although we initially identified CTNIPs based on their transcriptional up-regulation upon elicitor treatment, we have so far no direct evidence that they are involved in immunity, as, for example, we did not observe any difference in susceptibility of *hsl3* and *ctnip* mutant plants to the hypovirulent *Pseudomonas syringae* pv *tomato* DC3000 *ΔAvrPto/ΔAvrPtoB* strain upon spray infection (*Figure 3—figure supplement 5*). Although further investigation is required to test additional pathogens and conditions, it is also plausible that CTNIPs are involved in the regulation of plant growth during plant-microbe interactions – a hypothesis that needs to be tested in the future.

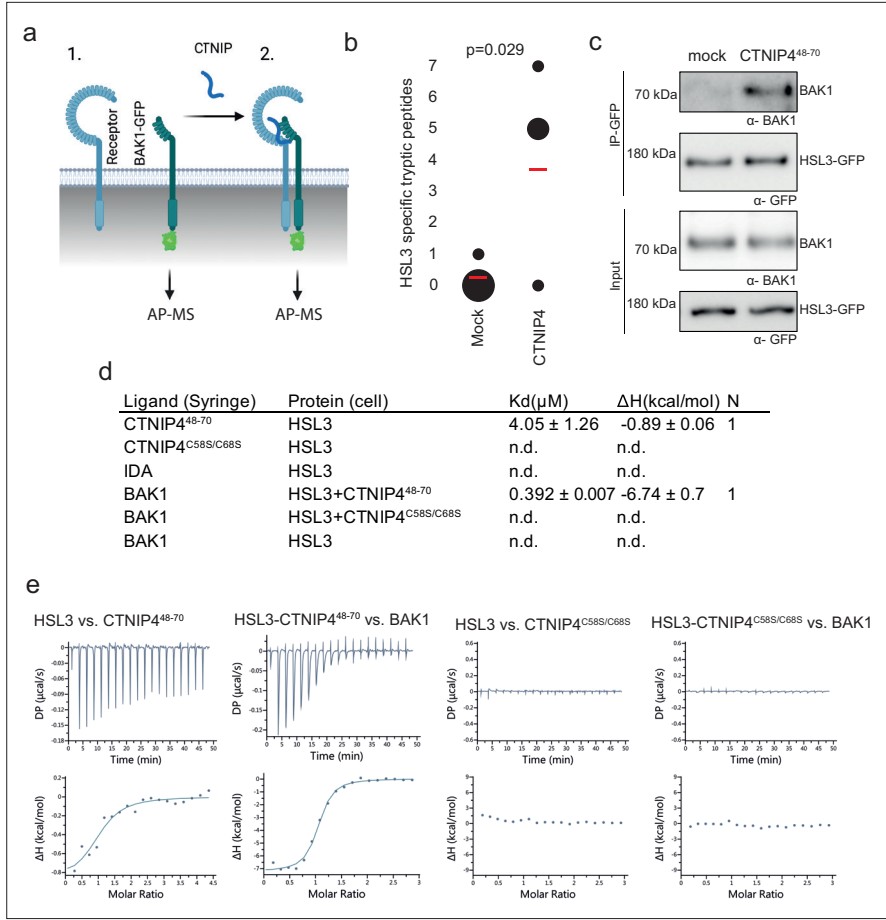

**Figure 2.** HAESA-LIKE 3 (HSL3) forms a CTNIP-induced receptor complex with BAK1. (**a**) Schematic representation of BAK1-GFP immunoprecipitation in the (1) absence or (2) presence of CTNIP4 treatment to identify protein associations induced by CTNIP. Figure generated using Biorender. (**b**) HSL3-specific spectral counts identified in four independent biological replicates where BAK1-GFP was pulled down in the presence or absence of 1 μM CTNIP4 treatment. Circle diameter is proportional to the number of replicates. Red lines indicate the mean spectral counts for each treatment. p-Values indicate significance relative to the untreated control in a two-tailed t-test. (**c**) Affinity purification of BAK1 with HSL3-GFP from HSL3-GFP seedlings treated with 1 μM CTNIP4[48-70] or water for 10 min. Western blots were probed with antibodies α-GFP and α-BAK1. This experiment was repeated three times with similar results. (**d**) Isothermal titration calorimetry (ITC) summary table of HSL3 vs. CTNP4[48-70], CTNIP4[C58S/C68S] and INFLORESCENCE DEFICIENT IN ABSCISSION (IDA) peptides, and contribution of the BAK1 co-receptor to the ternary complex formation. $K_d$, (dissociation constant) indicates the binding affinity between the two molecules considered (nM). The N indicates the reaction stoichiometry (N=1 for a 1:1 interaction). The values indicated in the table are the mean ± SEM of two independent experiments. (**e**) ITC experiments of HSL3 vs. CTNIP4 and CTNIP4[C58S/C68S], in the absence and presence of the co-receptor BAK1. GFP, green fluorescent protein.

The online version of this article includes the following source data and figure supplement(s) for figure 2:

**Source data 1.** HSL3-specific spectral counts.

**Source data 2.** Westernblots showing affinity purification of BAK1 with HSL3-GFP.

**Figure supplement 1.** Phylogeny of *Arabidopsis* leucine-rich repeat receptor kinase (LRR-RK) subfamily XI.

**Figure supplement 1—source data 1.** Flg22-induced expression data of LRR-RK subfamily XI.

**Figure supplement 2.** Isothermal titration calorimetry (ITC) independent experiments and purification of HAESA-LIKE 3 (HSL3) and BAK1 used in the binding experiments.

**Figure supplement 3.** Structural comparison of the binding pockets between the receptors HAESA and HAESA-LIKE 3 (HSL3).

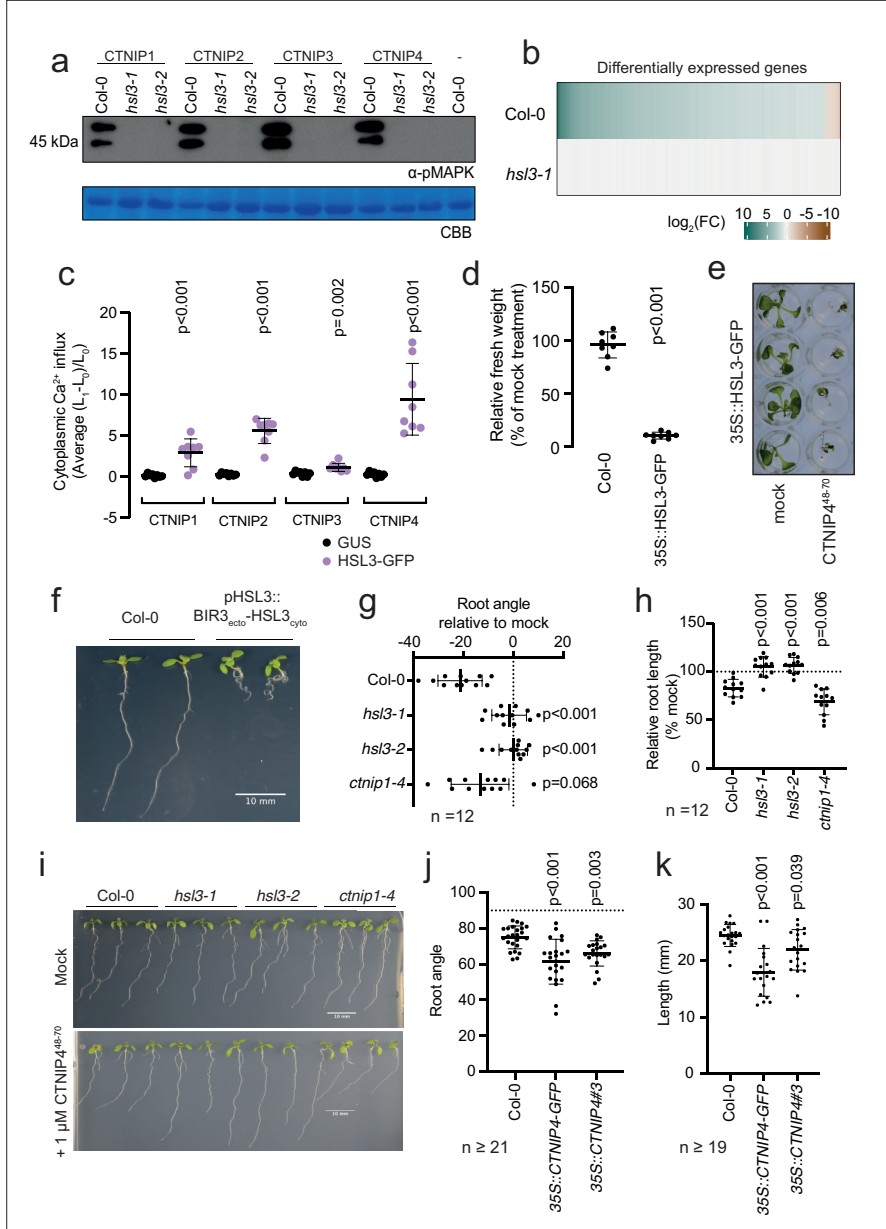

**Figure 3.** HAESA-LIKE 3 (HSL3) is strictly required for CTNIP perception and growth regulation. (**a**) Western blot using α-p42/p44-ERK recognising phosphorylated MAP kinases in seedlings treated with 100 nM CTNIPs or mock for 15 min. The membrane was stained with Coomassie brilliant blue (CBB), as a loading control. (**b**) Heat map showing all significantly differentially expressed genes (p<0.05, |log$_2$(FC)|>1) in *Arabidopsis* wild-type (WT) or *hsl3-1* seedlings treated with or without 100 nM CTNIP4$^{48-70}$ for 30 min relative to a mock control. (**c**) Mean relative cytoplasmic Ca$^{2+}$ influx in leaf disks of *Nicotiana benthamiana* transiently expressing the defined constructs induced by 1 µM CTNIP or mock application (n = 8 leaf disks). A line represents mean; error bars represent SD. p-Values indicate significance relative to the GUS-transformed control in a Dunnett's multiple comparison test following one-way ANOVA. (**d**) Fresh weight of 14-day-old seedlings grown in the presence of 500 nM CTNIP4$^{48-70}$ for 10 days relative to mock (n = 8 seedlings). A line represents mean; error bars represent SD. p-Values indicate significance relative to the WT control in a two-tailed t-test. (**e**) Representative images of (**d**). (**f,i**) Nine-day-old vertically grown *Arabidopsis* seedlings on 1/2 Murashige and Skoog (MS) agar medium with 1% sucrose. Pictures were taken from the front of the plate. (**g–h,j–k**) Root parameters were quantified from the base of the hypocotyl to the root tip using ImageJ. (**g**) Root angle is shown relative to mock. Negative values indicate leftward root skewing. (**j**) Absolute root angle with 90° representing the gravity vector. Angles < 90° represent skewing to the left. A line represents mean; error bars represent SD. p-Values indicate significance relative to the WT control in a Dunnett's

*Figure 3 continued on next page*

*Figure 3 continued*

multiple comparison test following one-way ANOVA. All experiments were repeated and analysed three times with similar results.

The online version of this article includes the following source data and figure supplement(s) for figure 3:

**Source data 1.** HSL3 is strictly required for CTNIP perception and growth regulation.

**Source data 2.** HSL3-dependency of CTNIP-induced MAPK phosphorylation.

**Figure supplement 1.** Genetic characterisation of *hsl3* mutants.

**Figure supplement 1—source data 1.** Genetic characterisation of *hsl3* mutants.

**Figure supplement 2.** CTNIP-induced seedling growth inhibition.

**Figure supplement 2—source data 1.** CTNIP-induced seedling growth inhibition.

**Figure supplement 3.** Correlation of CTNIP4-induced transcriptomic response with that of elicitors at 30min.

**Figure supplement 3—source data 1.** Correlation of CTNIP4-induced transcriptomic response with that of elicitors at 30min.

**Figure supplement 4.** Characterisation of CTNIP and chimeric receptor lines.

**Figure supplement 4—source data 1.** Characterisation of CTNIP and chimeric receptor lines.

**Figure supplement 5.** *HSL3-CTNIP* mutants do not show altered resistance to *Pseudomonas syringae* pv tomato DC3000 *ΔAvrPto/ΔAvrPtoB*.

**Figure supplement 5—source data 1.** *HSL3-CTNIP* mutants do not show altered resistance to *Pseudomonas syringae* pv tomato DC3000 *ΔAvrPto/ΔAvrPtoB*.

Recent phylogenetic analyses indicate that HSL3 is conserved in angiosperms (*Figure 4a*; *Figure 4—figure supplement 1*; *Supplementary file 16*; *Supplementary file 17*; *Supplementary file 18*; *Supplementary file 19*; *Supplementary file 20*; *Furumizu et al., 2021*; *Man et al., 2020*). Having defined HSL3 as the only CTNIP receptor, we wondered whether CTNIPs were equally conserved. CTNIPs were identified in *Amborella*, eudicots and monocots, excluding Poaceae (*Figure 4b–d*).

Given the conservation of the HSL3-CTNIP signalling module, the lack of *At*CTNIP4 responses in *N. benthamiana* suggests a co-evolution of ligand-receptor specificity, as for example previously proposed for PLANT ELICITOR PEPTIDE (PEP)-PEP RECEPTOR (PEPR) pairs (*Huffaker, 2015*; *Lori et al., 2015*). Accordingly, transient expression of *Medicago truncatula* HSL3 (*MtHSL3*; *Medtr3g110450*) in *N. benthamiana* only induced a cytoplasmic calcium influx upon treatment with a conspecific CTNIP (*Mt*CTNIP, Medtr1g044470) (*Figure 4e*). Whilst the receptor and the peptides appear conserved, we cannot conclude that they are functional orthologs. Further work is required to establish the roles and specificity of HSL3 and CTNIPs in diverse lineages.

Our phylogenetic analysis surprisingly revealed that no clear CTNIP could be found in Poaceae genomes (*Figure 4b–d*). Interestingly, this absence coincides with an expansion of HSL3 paralogs within these genomes (*Figure 4b*). We can speculate that the HSL3-CTNIP signalling module may have diverged considerably in this lineage, this may be reflected in the longer branch lengths observed in the HSL3 phylogeny, especially within the CTNIP-binding LRR domain (*Figure 4—figure supplement 1*). Interestingly, over 40% of the CTNIPs identified were unannotated, including all monocot CTNIPs (*Figure 4b*), highlighting how genome annotation still represents a significant challenge in the characterisation of signalling peptides.

## Conclusion

Here, we identified CTNIPs as a novel family of stress-induced signalling peptide. Using affinity purification and mass spectrometry based on ligand-induced association with the BAK1 co-receptor, we identified the LRR-RK HSL3 as the CTNIP receptor. CTNIPs directly bind the HSL3 ectodomain to promote BAK1 recruitment, and HSL3 is necessary and sufficient to confer CTNIP perception. Notably, HSL3 has been independently identified as the CTNIP receptor (there named SMALL PHYTOCYTO-KINES REGULATING DEFENSE AND WATER LOSS; SCREW) (*Liu et al., 2022*). This signalling module has been conserved for more than 180 million years (*Furumizu et al., 2021*; *Kumar et al., 2017*); however, its physiological role requires additional elucidation. HSL3 has recently been shown to play a role in regulating drought and disease resistance implicating HSL3 in multiple stress responses (*Liu*

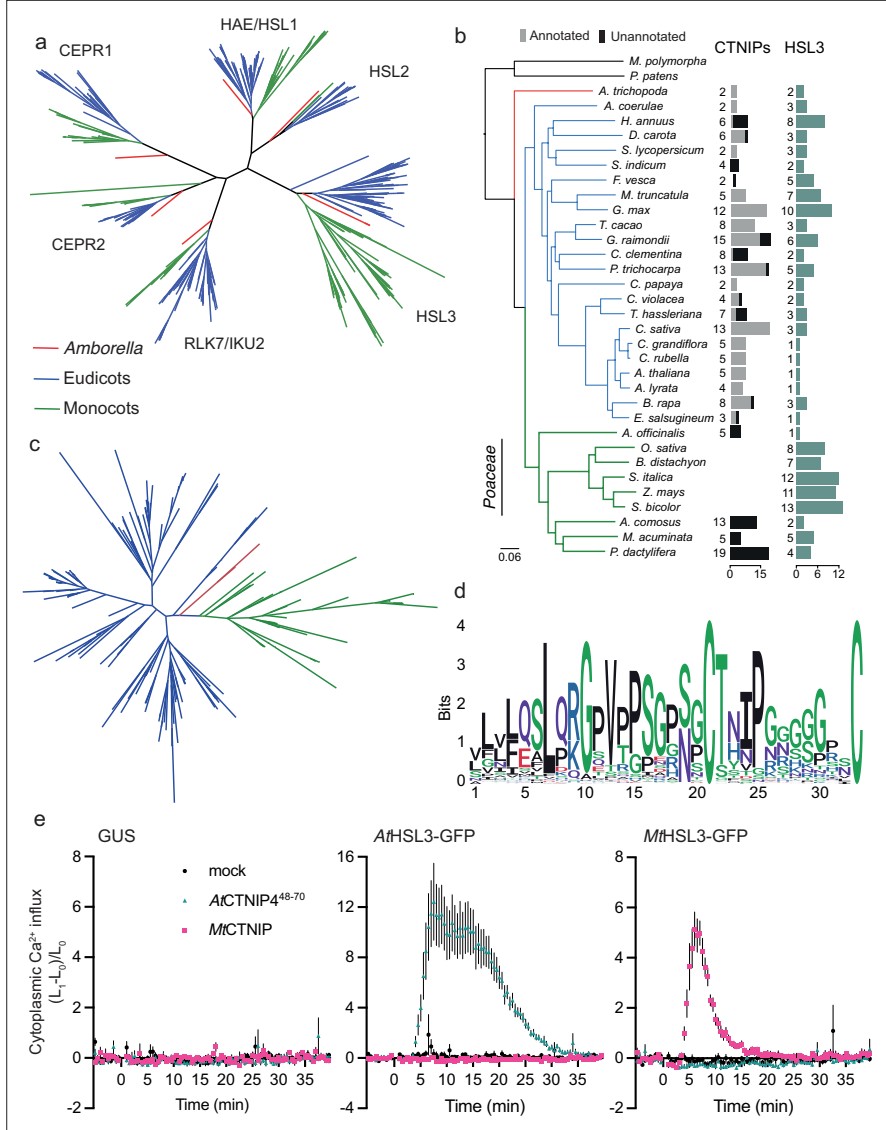

**Figure 4.** The HSL3-CTNIP signalling module predates extant angiosperms. (**a**) Phylogeny of the full-length amino acid sequences of HAE/HSL/CEPR/RLK7/IKU2 clade of receptor kinases. Eudicot sequences are indicated in blue, monocot sequences in green, and *Amborella* sequences in red. Clades are named based upon the *Arabidopsis* genes. Alignment shown in *Supplementary file 14*. Further details of species, sequence identification, alignment, and phylogeny generation are described in the Materials and methods. (**b**) Species tree with number of CTNIP and HSL3 orthologs identified. Annotated CTNIPs are shown in grey whilst unannotated CTNIPs are shown in black. Sequences are shown in *Supplementary files 10 and 16*. (**c**) Phylogeny of the full-length amino acid sequences of CTNIPs. Eudicot sequences are indicated in blue, monocot sequences in green, and *Amborella* sequences in red. Sequences shown in *Supplementary file 10*. Further details of species, sequence identification, alignment, and phylogeny generation are described in the Materials and methods. (**d**) Sequence logo generated from CTNIP alignment from (**c**) using the R-package ggseqlogo. Amino acids are coloured based on their biochemical properties: red = acidic; blue = basic; black = hydrophobic; purple = neutral, and green = polar. (**e**) Cytoplasmic calcium influx measured after treatment with 1 µM CTNIP in *p35S::AEQUORIN Nicotiana benthamiana* leaf disks transiently expressing the defined construct, relative to pre-treatment (n = 8 leaf disks). Points represent mean; error bars represent SEM. Experiments were repeated and analysed three times with similar results. HSL3, HAESA-LIKE 3.

The online version of this article includes the following source data and figure supplement(s) for figure 4:

**Source data 1.** The HSL3-CTNIP signalling module predates extant angiosperms.

**Figure supplement 1.** HAE/HSL/CEPR/RLK7/IKU2 clade phylogenies.

*et al., 2020; Liu et al., 2022*). Deorphanising HSL3 makes LRR-RK subfamily XI an exciting tool to understand receptor-ligand co-evolution and recognition specificity.

## Materials and methods

**Key resources table**

| Reagent type (species) or resource | Designation | Source or reference | Identifiers | Additional information |
|---|---|---|---|---|
| Gene (*Arabidopsis thaliana*) | HSL3 | ARAPORT11 | AT5G25930, Q9XGZ2_ARATH | |
| Gene (*Arabidopsis thaliana*) | CTNIP1 | ARAPORT11 | AT1G06135, Q8LCX3_ARATH | |
| Gene (*Arabidopsis thaliana*) | CTNIP2 | ARAPORT11 | AT1G06137, F4IBZ9_ARATH | |
| Gene (*Arabidopsis thaliana*) | CTNIP3 | ARAPORT11 | AT2G31335, Q1G3B9_ARATH | |
| Gene (*Arabidopsis thaliana*) | CTNIP4 | ARAPORT11 | AT2G31345, Q8L9Z1_ARATH | |
| Gene (*Arabidopsis thaliana*) | CTNIP5 | ARAPORT11 | AT2G23123, A0A1I9LM80_ARATH | |
| Gene (*Medicago truncatula*) | MtHSL3 | Mt4.0 | G7J3I8_MEDTR, MTR_3g110450, MtrunA17_Chr3g0140551 | |
| Gene (*Medicago truncatula*) | MtCTNIP | Mt4.0 | G7I613_MEDTR, MTR_1g044470, MtrunA17_Chr1g0168241 | |
| Genetic reagent (*Arabidopsis thaliana*) | *hsl3-1* | euNASC | salk_207895 | |
| Genetic reagent (*Arabidopsis thaliana*) | *hsl3-2* | euNASC | wiscdslox450b04 | |
| Genetic reagent (*Arabidopsis thaliana*) | *bak1-4/pBAK1::BAK1-GFP* | https://doi.org/10.1105/tpc.111.090779 | *bak1-4/pBAK1::BAK1-GFP* | |
| Genetic reagent (*Arabidopsis thaliana*) | *bak1-5* | https://doi.org/10.1371/journal.pgen.1002046 | *bak1-5* | |
| Genetic reagent (*Arabidopsis thaliana*) | *p35S::AEQUORIN* | https://doi.org/10.1038/352524a0 | | |
| Genetic reagent (*Nicotiana benthamiana*) | *p35S::AEQUORIN* | https://doi.org/10.1104/pp.110.171249 | | |
| Genetic reagent (*Arabidopsis thaliana*) | 35S::HSL3-GFP | This paper | | *Figure 3* and Materials and methods |
| Genetic reagent (*Arabidopsis thaliana*) | pHSL3::BIR3-HSL3-FLAG | This paper | | Chimera created using the method from https://doi.org/10.1105/tpc.20.00138 |
| Cell line (*Trichoplusia ni*) | Tnao38 | https://doi.org/10.1186/1472-6750-12-12 | | Cell line maintained in J Santiago lab |
| Antibody | Anti-BAK1 (rabbit polyclonal) | https://doi.org/10.1105/tpc.111.084301 | | WB (1:2000) |
| Antibody | Anti-GFP (HRP-conjugated mouse monoclonal) | Santa Cruz | sc-9996 | WB (1:5000) |
| Antibody | Anti-pMAPK (rabbit polyclonal) | Cell Signaling | p44/42 MAPK (Erk1/2) antibody #9,102 | WB (1:4000) |
| Recombinant DNA reagent | 35S::HSL3-GFP (plasmid) | This paper | | *Figure 3* |

*Continued on next page*

*Continued*

| Reagent type (species) or resource | Designation | Source or reference | Identifiers | Additional information |
|---|---|---|---|---|
| Recombinant DNA reagent | pHSL3::BIR3-HSL3-FLAG (plasmid) | This paper | | Used to generate transgenic plants in *Figure 3* |
| Recombinant DNA reagent | pHSL3::LTI6B-Citrine (plasmid) | This paper | | Used to generate transgenic plants in *Figure 3—figure supplement 4* |
| Recombinant DNA reagent | pHSL3::BIR3$^{F146A,R170A}$-HSL3-Citrine (plasmid) | This paper | | Used to generate transgenic plants in *Figure 3—figure supplement 4* |
| Recombinant DNA reagent | pHSL3::BIR3-HSL3-CITRINE (plasmid) | This paper | | Used to generate transgenic plants in *Figure 3—figure supplement 4* |
| Commercial assay or kit | GFP-Trap | Chromotek | Cat. #: gta-20 | |
| Software, algorithm | GraphPad | GraphPad software | | |

## Plant material and growth conditions

*Arabidopsis* plants for ROS burst assays were grown in individual pots at 21°C with a 10 hr photo-period. Seeds grown on plates were surface sterilised using chlorine gas for 5–6 hr and sown on 1/2 Murashige and Skoog (MS) media, 1% sucrose, and 0.8% agar and stratified at 4°C for 2–3 days. Plates were then transferred to 22°C under a 16 hr photoperiod. For root growth assays, plates were placed in an upright position under a 10° angle relative to the direction of gravity and images were taken 9 days later (*Van der Does et al., 2017*). For T1 selection sterilised seeds were sown on plates containing 100 µg/mL carbenicillin and 15 µg/mL glufosinate-ammonium prior to stratification, and grown for 4 days until resistant plants could be selected. Seedlings were then transferred to fresh plated and grown for a further 5 days before imaging. *N. benthamiana* plants were grown on peat-based media at 24°C, with 16 hr photoperiod.

Aequorin lines of *Arabidopsis* and *N. benthamiana* were described previously (*Knight et al., 1991*; *Segonzac et al., 2011*). *Hsl3* mutants have been previously described (*hsl3-1* (salk_207895), *hsl3-2* (wiscdslox450b04)) and were obtained from the Eurasian *Arabidopsis* Stock Centre (uNASC) (*Hou et al., 2014*; *Liu et al., 2020*). *bak1-4/pBAK1::BAK1-GFP* and *bak1-5* lines have also been described previously (*Ntoukakis et al., 2011*; *Schwessinger et al., 2011*).

## Initial identification of CTNIPs

Expression data were taken from *Bjornson et al., 2021*. Transcripts for all Araport11 gene models encoding proteins less than 150 amino acids were ranked based on flg22-induced transcript accumulation at 90 min (*Supplementary file 1*). Signal peptides were predicted using SignalP5.0. Reciprocal BLAST was used to identify similar sequences. CTNIP1, CTNIP2, and CTNIP5 were identified by BLAST against the Araport11 proteome.

## Synthetic peptides

Initially, synthetic peptides were ordered based on the full-length peptides with the predicted signal peptide removed. Subsequently, we divided the initial peptide into two fragments (CTNIP4$^{27-48}$ and CTNIP4$^{48-70}$; *Figure 1—figure supplement 2*). Whilst the peptide CTNIP4$^{48-70}$ was sufficient for binding and to induce responses, we do not currently have evidence to suggest whether this corresponds to the peptide produced in planta or whether the peptide produced in planta is post-translationally modified (*Matsubayashi, 2014*). This may impact the bioactivity of the peptide. All synthetic peptides were ordered at >80% purity from either Ezbiolabs or GenScript. Sequences of all peptides can be found in *Supplementary file 6*.

## Alignment and phylogeny of *Arabidopsis* CTNIPs and LRR-RK subfamily XI

Full-length protein sequences were aligned using MUSCLE (*Edgar, 2004*) and a phylogeny was inferred using the maximum-likelihood method and JTT matrix-based model conducted in MEGAX (*Kumar et al., 2018*). 1000 bootstraps were performed. Trees were visualised in iTOL (*Letunic and Bork, 2019*). The sequence logo was generated using WebLogo3 (*Crooks et al., 2004*).

## In-planta expression

For overexpression of *At*HSL3-GFP and *MtHSL3* in *N. benthamiana* and *Arabidopsis*, the genomic DNA sequence was amplified from *Arabidopsis* ecotype Columbia and *M. truncatula* ecotype A11, domesticated and directly ligated into pICSL86977 downstream of a 35S promoter and with an in-frame C-terminal GFP tag.

Fragments for the *pHSL3::BIR3ecto-HSL3cyto-FLAG*, *pHSL3::BIR3ecto-HSL3cyto-citrine*, *pHSL3::BIR3$^{F146A/R170A}$ecto-HSL3cyto-FLAG,* and *pHSL3::LTI6B-citrine* construct were amplified from genomic DNA using the indicated primers or synthesised by Twist Bioscience and ligated into pICSL86955 (*Supplementary file 5*). Fragments were designed according to *Hohmann et al., 2020*. Clones were verified by Sanger sequencing.

## CRISPR-Cas9 mutagenesis

CRISPR-Cas9-induced mutagenesis was performed as described by *Castel et al., 2019*. The *RPS5a* promoter drove Cas9 expression and FASTred selection was used for positive and negative selection. Primers used to generate the vector can be found in *Supplementary file 5*. Mutants were screened by Sanger sequencing.

## ROS measurements

Leaf disks were harvested from 4-week-old *Arabidopsis* plants into white 96-well-plates (655075, Greiner Bio-One) containing 100 µL water using a 4 mm diameter biopsy punch (Integra Miltex). Leaf disks were rested overnight. Prior to ROS measurement, the water was removed and replaced with ROS assay solution (100 µM Luminol [123072, Merck], 20 µg/mL$^{-1}$ horseradish peroxidase [P6782, Merck]) with or without elicitors. Immediately after light emission was measured from the plate using a HIGH-RESOLUTION PHOTON COUNTING SYSTEM (HRPCS218, Photek) equipped with a 20 mm F1.8 EX DG ASPHERICAL RF WIDE LENS (Sigma Corp).

## Cytoplasmic calcium measurements

Seedlings were initially grown on 1/2 MS plates for 3 days before being transferred to 96-well plates (655075, Greiner Bio-One) in 100 µL liquid MS for 5 days. The evening before calcium measurements the liquid MS was replaced with 100 µL 20 µM coelenterazine (EC14031, Carbosynth) and the seedlings incubated in the dark overnight. The following morning the coelenterazine solution was replaced with 100 µL water and rested for a minimum of 30 min in the dark. Readings were taken in a VARIOSKAN MUTIPLATE READER (ThermoFisher) before and after adding 50 µL of 3×concentrated elicitor solution or mock. For each well readings were normalised to the average RLU value before elicitor addition (L$_0$).

## Seedling growth inhibition

Four-day-old seedlings growing on 1/2 MS plates were transferred into individual wells of a transparent 48-well tissue culture plate (Greiner Bio-One) containing 500 µL of liquid MS media with/without elicitor addition. The plates were returned to the growth conditions for an additional 10 days before seedlings were blot-dried and weighed.

## *Pseudomonas* infection

*P. syringae* pv tomato DC3000 *ΔAvrPto/ΔAvrPto* was cultured overnight in liquid Kings B medium at 28°C. Cells were harvested by centrifugation, and pellets were resuspended in 10 mM MgCl$_2$ to OD$_{600}$=0.2. Immediately prior to spraying, 0.04% (v/v) Silwet L-77 was added. Bacteria were sprayed onto leaf surfaces until runoff and plants were maintained at high humidity for 3 days. Subsequently

samples were taken using a biopsy punch (4 mm diameter) to cut two leaf discs per leaf, three leaves per plant. Leaf discs were ground in 10 mM MgCl₂, diluted, and plated on L agar with appropriate selection. Plates were incubated at 28°C and colonies counted after ~1.5 days growth.

## Protein extraction and western blot

Two-week-old seedlings grown in liquid MS media (MAPK phosphorylation) or leaf disks from 4-week-old plants were flash-frozen in liquid nitrogen. Frozen plant tissue was ground in a Geno-grinder with 2 mm glass beads (1500 strokes/min, 1.5 min) prior to boiling in 2× Laemmli sample buffer (4% SDS, 20% glycerol, 10% 2-mercaptoethanol, 0.004% bromophenol blue, and 0.125 M Tris-HCl; 10 µL/mg tissue) for 10 min at 95°C. The samples were then spun at 13,000 rcf for 5 min prior to loading and running on SDS-PAGE gels. Proteins were transferred using semi-dry transfer onto PVDF membrane (ThermoFisher), blocked in 5% (w/v) bovine serum albumin prior to incubation with appropriate antibodies α-pMAPK (p44/42 MAPK [Erk1/2] antibody #9102; 1:4000); α-FLAG-HRP (A8592, Merck; 1:5000); and α-rabbit-HRP (A-0545, Merck; 1:10,000). Western blots were imaged with a LAS 4000 IMAGEQUANT SYSTEM (GE Healthcare) or on X-ray film before being developed. Staining of the blotted membrane with Coomassie brilliant blue was used to confirm loading.

## Affinity purification and western blotting

All steps involving the protein extract and subsequent protein isolation were carried out on ice or at 4°C and all buffers and tubes were pre-cooled.

Seeds were sown on 1/2 MS agar and stratified for 3 days as described above. When seedlings had germinated, they were transferred 6 seedlings per well into 6-well plates containing 5 mL of liquid MS media and grown for a further 12 days. Seedlings were then transferred into MS media either with or without CTNIP4 addition, vacuum infiltrated for 2 min, and left in the solution for a further 10 min. Seedlings were rapidly dried and flash frozen in liquid nitrogen and ground. Proteins were extracted using by addition of ~2:1 extraction buffer (50 mM Tris pH 7.5, 150 mM NaCl, 2.5 mM EDTA, 10% glycerol, 1% IGEPAL, 5 mM DTT, 1% plant protease inhibitor cocktail [P9599, Sigma]): ground tissue (v/v). Proteins were solubilised at 4°C with gentle agitation for 30 min before filtering through Miracloth. The filtrate was centrifuged at 30,000 rcf for 30 min at 4°C. Protein concentrations were normalised using Bradford assay. An input sample was taken. To each 15 mL of protein extract 40 µL of GFP-TRAP AGAROSE BEADS (50% slurry, ChromoTek) washed in extraction buffer were added and incubated with gentle agitation for 4 hr at 4°C. Beads were harvested by centrifugation at 1500× *g* for 2 min and washed three times in extraction buffer. Beads were then resuspended in 50 µL of 1.5× elution (NuPage) buffer and incubated at 80°C for 8 min. Samples were subsequently used from MS analysis or western blotting.

Western blotting was performed as described previously: α-BAK1 (*Roux et al., 2011*; 1:2000), α-GFP-HRP (sc-9996, Santa Cruz; 1:5000), and α-rabbit-HRP (A-0545, Merck; 1:10,000).

## Sample preparation for mass spectrometry

Affinity-purified protein samples were run approximately 1 cm into an SDS-PAGE gel. This portion of the gel was then excised, cut into smaller pieces, and washed three times with acetonitrile (LC-MS-Grade):ammonium bicarbonate (50 mM), pH 8.0 (1:1, v/v), 30 min each, followed by dehydration in acetonitrile, 10 min. Gel pieces were then reduced with 10 mM DTT for 30 min at 45°C followed by alkylation with 55 mM iodoacetamide for 20 min at room temperature, and a further three washes with acetonitrile:ammonium, 30 min each. Gel pieces were dehydrated again with acetonitrile before rehydration with 40 µL trypsin (Pierce Trypsin Protease, MS-Grade, catalog no. 90058) working solution (100 ng trypsin in 50 mM ammonium bicarbonate, 5% (v/v) acetonitrile). Where required, gel pieces were covered with 50 mM ammonium bicarbonate to a final volume before incubation at 37°C overnight. Tryptic peptides were extracted from the gel pieces three times in an equal volume of 50% acetonitrile, 5% formic acid (Pierce LC-MS-Grade, catalog no. 85178), 30 min each. Extracted peptides were dried in a speed-vac and resuspended in 2% acetonitrile/0.2% trifluoroacetic acid (Merck, catalog no. 302031). A total of four biological replicates for each sample type were submitted.

## LC-MS/MS analysis

Approximately 35% of each sample was analysed using an Orbitrap Fusion Tribrid Mass Spectrometer (Thermo Fisher Scientific) coupled to a U3000 nano-UPLC (Thermo Fisher Scientific). The dissolved peptides were injected onto a reverse phase trap column NanoEase m/z Symmetry C18, beads diameter 5 µm, inner diameter 180 µm × 20 mm length (Waters). The column was operated at the flow rate 20 µL/min in 2% acetonitrile, 0.05% TFA, after 2.5 min the trap column was connected to the analytical column NanoEase m/z HSS C18 T3 Column, beads diameter 1.8 µm, inner diameter 75 µm × 250 mm length (Waters). The column was equilibrated with 3% B (B: 80% acetonitrile in 0.05% formic acid [FA], A: 0.1% FA) before subsequent elution with the following steps of a linear gradient: 2.5 min 3% B, 5 min 6.3% B, 13 min 12.5% B, 50 min 42.5% B, 58 min 50% B, 61 min 65% B, 63 min 99% B, 66 min 99% B, 67 min 3% B, 90 min 3% B. The flow rate was set to 200 nL/min. The mass spectrometer was operated in positive ion mode with nano-electrospray ion source. Molecular ions were generated by applying voltage +2.2 kV to a conductive union coupling the column outlet with fused silica PicoTip emitter, ID 10 µm (New Objective, Inc) and the ion transfer capillary temperature was set to 275°C. The mass spectrometer was operated in data-dependent mode using a full scan, m/z range 300–1800, nominal resolution of 120,000, target value $1\times10^6$, followed by MS/MS scans of the 40 most abundant ions. MS/MS spectra were acquired using normalised collision energy of 30%, isolation width of 1.6 m/z, resolution of 120,000, and a target value set to $1\times10^5$. Precursor ions with charge states 2–7 were selected for fragmentation and put on a dynamic exclusion list for 30 s. The minimum automatic gain control target was set to $5\times10^3$ and intensity threshold was calculated to be $4.8\times10^4$. The peptide match feature was set to the preferred mode and the feature to exclude isotopes was enabled.

## Data processing and peptide identification

Peak lists in the form of Mascot generic files were prepared from raw data files using MS Convert (Proteowizard) and sent to a peptide search on Mascot server v2.7 using Mascot Daemon (Matrix Science, Ltd) against an in-house contaminants database and the Araport 11 protein database. Tryptic peptides with up to one possible mis-cleavage and charge states +2, +3 were allowed in the search. The following peptide modifications were included in the search: carbamidomethylated cysteine (fixed) and oxidised methionine (variable). Data were searched with a monoisotopic precursor and fragment ion mass tolerance 10 ppm and 0.8 Da, respectively. Decoy database was used to validate peptide sequence matches. Mascot results were combined in Scaffold v4.4.0 (Proteome Software Inc) and peptide and protein identifications accepted if peptide probability and protein threshold was ≥95.0% and 99%, respectively. Under these conditions the false discovery rate was 0.06%. Data was then exported to Excel (Microsoft) for further processing. Proteins were accepted if identified by at least two peptides and present in two or more biological replicates. Spectral counts from the four biological replicates were summed and used to derive a ratio of CTNIP4 treatment:mock treatment. The mass spectrometry proteomics data have been deposited to the ProteomeXchange Consortium via the PRIDE (*Perez-Riverol et al., 2019*) partner repository with the dataset identifier PXD029264 and 10.6019/PXD029264.

## Transient expression in *N. benthamiana*

*Agrobacterium tumefaciens* strain GV3101 transformed with the appropriate construct were grown overnight in L-media and spun-down. The bacteria were resuspended in 10 mM $MgCl_2$ and adjusted to $OD_{600}=0.2$ prior to infiltration into the youngest fully expanded leaves of 3-week-old plants. Leaf disks were collected 24 hr later, and calcium assays were performed as described for seedlings with leaf disks being floated overnight in the dark in 20 µM coelenterazine (EC14031, Carbosynth).

## Protein expression and purification

The ectodomains expressed and purified were coded from *Arabidopsis* genes *HSL3* (22–627, AT5G25930) and *BAK1* (residues 20–637, AT4G33430). Codon-optimised synthetic genes were cloned into a modified pFastBac vector (Geneva Biotech) vector, providing a TEV (tobacco etch virus) protease cleavable C-terminal StrepII-9xHis tag. Expression of HSL3 and BAK1 was driven by the signal peptides 30 K (*Futatsumori-Sugai and Tsumoto, 2010*) or *Drosophila* BiP (*Smakowska-Luzan et al., 2018*), respectively. The baculovirus were generated in DH10 cells and *Spodoptera frugiperda* Sf9 cells were used for viral amplification. For protein expression *Trichoplusia ni* Tnao38

cells (*Hashimoto et al., 2012*) were infected with HSL3 and BAK1 viruses with a multiplicity of infection of 3. The cells were grown 1 day at 28°C and 2 days at 22°C at 110 rpm. The secreted proteins were purified separately by sequential $Ni^{2+}$ (HisTrap excel, GE Healthcare, equilibrated in 25 mM $KP_i$ pH 7.8 and 500 mM NaCl) and StrepII (Strep-Tactin Superflow high-capacity, [IBA, Germany] equilibrated in 25 mM Tris pH 8.0, 250 mM NaCl, 1 mM EDTA) affinity chromatography. Recombinant Strep-tagged TEV protease was used in 1:50 ratio to remove the affinity tags. The cleaved tag and the protease were separated from the protein ectodomains by $Ni^{2+}$ affinity chromatography. Proteins were further purified by size exclusion chromatography on a Superdex 200 Increase 10/300 GL column (GE Healthcare) equilibrated in 20 mM citric acid pH 5.0, 150 mM NaCl. Peak fractions containing the complex were concentrated using Amicon Ultra concentrators (Millipore, MWCO 10,000 for BAK1 and 30,000 for HSL3). Proteins were analysed for purity and structural integrity by SDS-PAGE.

## Isothermal titration calorimetry

A MicroCal PEAQ-ITC (Malvern Instruments) was used to performing the ITC-binding assays. Experiments were performed at 25°C with a 200 µL standard cell and a 40 µL titration syringe. HSL3 and BAK1 proteins were gel-filtrated into pH 5 ITC buffer (20 mM sodium citrate pH 5.0, 150 mM NaCl). Protein concentrations for HSL3 and BAK1 were calculated using their molar extinction coefficient and a calculated molecular weight of ~75,000 for HSL3 and ~25,000 Da for BAK1. Experiments were performed with 20 µM of HSL3 protein in the cell and between 200 and 450 µM of indicated peptide ligand in the syringe, following an injection pattern of 2 µL at 150 s intervals and 500 rpm stirring speed. The BAK1 vs. HSL3-peptide experiments were performed by titrating 100 µM of BAK1 in the cell, using the same injection pattern. ITC data were corrected for the heat of dilution by subtracting the mixing enthalpies for titrant solution injections into protein-free ITC buffer. Experiments were done in replicates and data were analysed using the MicroCal PEAQ-ITC Analysis Software provided by the manufacturer. All ITC runs used for data analysis had an N ranging from 0.7 to 1.3. The N values were fitted to 1 in the analysis.

## RNA sequencing and qRT-PCR

Two 3-day-old seedlings per well were transferred into transparent 24-well plates (Grenier Bio-One) containing 1 mL liquid MS media, sealed with porous tape and grown for a further 9 days. For qRT-PCR, seedlings were harvested at this point. For RNA sequencing experiments media was then exchanged for 500 µL fresh MS media and left overnight. In the morning a further 480 µL of fresh media was added; 9.5 hr later 20 µL treatment/mock was added and seedlings were harvested after 30 min. All seedlings were ground in liquid nitrogen.

Total RNA was extracted using Trizol reagent (Merck) according to the manufacturer's instructions and DNAase/RNA cleanup treatment was performed using the Rneasy kit (Qiagen). RNA sequencing was performed by Novogene. The RNA sequencing datasets generated and analysed in the current study have been deposited in the ArrayExpress database at EMBL-EBI (https://www.ebi.ac.uk/arrayexpress/) under accession number E-MTAB-11093. qRT-PCR was performed on cDNA synthesised using The RevertAid first strand cDNA synthesis kit (ThermoFisher) according to the manufacturer's instructions. cDNA was amplified by quantitative PCR using SYBR Green JumpStart Taq ReadyMix (Roche) and the CFX96 Real-Time PCR Detection System (Bio-Rad Laboratories, Hercules, CA).

The read data were analysed using FastQC, trimmed using trimmomatic (*Bolger et al., 2014*) and mapped to the *Arabidopsis* TAIR10 genome via TopHat2 (*Andrews et al., 2015*; *Kim et al., 2013*). The mapped reads were assigned to genes by featureCounts from package Rsubread in R (*Liao et al., 2019*), and differential expression analysis was performed using DESeq2 with ashr L2FC shrinkage (*Love et al., 2014*; *Stephens, 2017*). Changes in gene expression were visualised using the R package ComplexHeatmap (*Gu et al., 2016*).

## GO enrichment

GO term enrichment was calculated using the R package topGO (*Alexa and Rahnenfuhrer, 2021*), with arguments method = weight.01 and statistic = Fisher.

## Correlation of expression

Pairwise comparisons of gene expression differences ($\log_2$(FC)) was performed in R using the rcorr function from package Hmisc (*Harrell, 2021*), type = Spearman, and correlations were plotted using corrplot (*Wei and Simko, 2021*).

## Genome data retrieval

Whole genome sequences and protein sequences were retrieved from Ensembl (release 50), Phytozome (version 13), NCBI, and marchantia.info. Species and individual assembly versions are listed in *Supplementary file 7* (SI_table_species_data.csv).

## CTNIP identification

### Peptide search

Protein sequences from all species were first filtered for a maximum length of 300 amino acids and merged into a single file. The initial set of CTNIP peptide sequences is given in *Supplementary file 8* (Initial_CTNIP_candidates.fasta). Additional candidates were searched with (1) jackhmmer (version 3.1b2, *Eddy, 2011*), (2) diamond (version 0.9.26, options -e 1e-8 -k 100, *Buchfink et al., 2015*), and (3) hmm profile search (3.1b2, *Wheeler and Eddy, 2013*). For the hmm profile search, the initial set of candidates and the candidates from the diamond search were aligned with muscle (v3.8.31, *Edgar, 2004*) to generate an hmm profile (hmmbuild) that was then used to search more candidates (hmmsearch). Candidates from all approaches were merged and grouped with a sequence similarity network. For this, sequences were matched to each other with diamond (options -e 0.01 k 100). The pairwise percent similarity scores above 20% were used to construct a network. The community structure of the network was resolved with a modularity optimisation algorithm (*Blondel et al., 2008*) implemented by the function cluster_louvain in the R package igraph (version 1.0.1, *Csardi and Nepusz, 2006*). Candidates within the same communities as the original candidate sequences were used as protein candidates.

### DNA search

To search novel peptides that were previously not annotated, we extracted all transcript sequences of the protein candidates and aligned them with muscle to generate an HMM profile (hmmbuild) that was used to search all genomes with nhmmer (3.1b2, *Wheeler and Eddy, 2013*). Candidate regions were filtered for already annotated genes and used as input to restrict de novo gene prediction with Augustus (version 3.3.3, *Stanke et al., 2008*). Finally, candidates from both, protein and DNA search, were merged to generate the final set of CTNIP candidates (*Supplementary file 9*, CTNIP_relaxed. align). This 'relaxed' set of candidates was further filtered for having two cysteines with a 9–11 amino acid spacing. Few candidates were also removed by a visual inspection of the alignment, resulting in the 'confident' CTNIP candidates (*Supplementary file 10*, CTNIP_confident.align). Phylogeny and clade identification was done with the 'relaxed' set of candidates using muscle and FastTree (version 2.1.11 SSE3, option -lg, *Price et al., 2010*) using an age cutoff of 9. The resulting phylogenetic tree was rooted using a similar sequence from *Marchantia polymorpha* (chr5:16052258–16053618) as outgroup with gotree (v0.4.2, *Lemoine and Gascuel, 2021*) and graphically represented using FigTree (v1.4.3, http://tree.bio.ed.ac.uk/software/figtree). The sequence logo was generated with the alignment of the 'confident' CTNIP candidates and the R-package ggseqlogo (v.1, *Wagih, 2017*). Amino acids with a low occurrence (i.e., seen in less than 5% of the peptides) were trimmed from the alignment to generate a gap-free logo.

## RK identification

Protein sequences from all species were first filtered for a minimum length of 500 amino acids and merged into a single file. The initial set of RK sequences was taken from the alignment provided by *Furumizu et al., 2021*, but with the outgroups removed (*Penium margaritaceum*, *Sinningia muscicola*, and *Mesotaenium endlicherianum*). The sequences are given in *Supplementary file 11* (Initial_RK_candidates.fasta). Sequences were aligned with muscle to build and search an HMM profile (hmmsearch options -E 1e-10 –incE 1e-10). Candidates were matched to each other with diamond (options -e 1e-11 –id 20 –query-cover 80). The pairwise percent similarity scores above 50% were used to construct a network and communities were defined as described as above. Likewise, only

candidates within the same communities as the original candidates were kept. Candidates were further filtered for the presence of an LRR and a kinase domain with hmmsearch (options -E 1e-5) and PFAMv33 (*Supplementary file 12*, *Supplementary file 13*; RECEPTOR.align). Phylogeny and clade identification was done with muscle and FastTree (option -lg) using an age cutoff of 5.5. The resulting phylogenetic tree was rooted using the sequences from *P. margaritaceum* as outgroup (*Furumizu et al., 2021*; *Supplementary file 14*, *Supplementary file 15*).

For the HSL3-phylogeny, we extracted the kinase domain and the ectodomain of the receptors (*Supplementary file 16*, *Supplementary file 17*, *Supplementary file 18*, *Supplementary file 19*, *Supplementary file 20*). The most likely kinase domain region of each candidate was identified with hmmer using the PFAM PF00069.26 motif. To extract the ectodomain, signal peptides were removed with signalp (version 5.0b, *Almagro Armenteros et al., 2019*). The remaining sequence was then segmented into intracellular, extracellular, and membrane-spanning using tmhmm (version 2.0, *Krogh et al., 2001*). The longest extracellular domain was taken as ectodomain.

### Phylogenetic tree of all species

The species tree was calculated using OrthoFinder (v2.5.4, *Emms and Kelly, 2019*) with all protein sequences of all plants species.

## Acknowledgements

We thank the John Innes Centre Horticultural Services for plant care, especially T Wells; M Smoker, J Taylor, and A Wawryk from the TSL Plant Transformation support group for plant transformation and all past and current members of the Zipfel and Santiago groups for technical help and fruitful discussions. N Talbot is acknowledged for hosting JR for part of this study. This work was supported by the European Research Council under the Grant Agreements no. 773153 and no. 716358 (grant 'IMMUNO-PEPTALK' to CZ and grant 'WallWatchers' to JS, respectively), The Gatsby Charitable Foundation (to CZ), the University of Zürich (to CZ), the Swiss National Science Foundation grants no. 31003A_182625 (to CZ) and no. 31003A_173101 (to JS), and the Fondation Philanthropique Famille Sandoz (to JS). MB was partially supported by the European Union's Horizon 2020 Research and Innovation Program under Marie Skłodowska-Curie Actions (grant agreement no. 703954).

## Additional information

### Funding

| Funder | Grant reference number | Author |
|---|---|---|
| H2020 European Research Council | 773153 | Cyril Zipfel |
| H2020 European Research Council | 716358 | Julia Santiago |
| The Gatsby Charitable Foundation | | Cyril Zipfel |
| Universität Zürich | | Cyril Zipfel |
| Swiss National Science Foundation | 31003A_182625 | Cyril Zipfel |
| Swiss National Science Foundation | 31003A_173101 | Julia Santiago |
| Fondation philanthropique Famille Sandoz | | Julia Santiago |
| H2020 Marie Skłodowska-Curie Actions | 703954 | Marta Bjornson |
| Biotechnology and Biological Sciences Research Council | BB/P012574/1 | Cyril Zipfel |

| Funder | Grant reference number | Author |
|---|---|---|

The funders had no role in study design, data collection and interpretation, or the decision to submit the work for publication.

## Author contributions
Jack Rhodes, Conceptualization, Data curation, Investigation, Visualization, Writing - original draft; Andra-Octavia Roman, Marc W Schmid, Data curation, Formal analysis, Investigation, Methodology, Visualization, Writing – review and editing; Marta Bjornson, Data curation, Formal analysis, Methodology, Visualization, Writing – review and editing; Benjamin Brandt, Investigation, Writing – review and editing; Paul Derbyshire, Data curation, Formal analysis, Investigation, Methodology, Writing – review and editing; Michele Wyler, Data curation, Formal analysis, Investigation; Frank LH Menke, Supervision, Writing – review and editing; Julia Santiago, Conceptualization, Funding acquisition, Methodology, Supervision, Writing – review and editing; Cyril Zipfel, Conceptualization, Funding acquisition, Project administration, Supervision, Writing – review and editing

## Author ORCIDs
Jack Rhodes http://orcid.org/0000-0002-3953-1648
Andra-Octavia Roman http://orcid.org/0000-0002-3037-3321
Marta Bjornson http://orcid.org/0000-0002-8275-4521
Benjamin Brandt http://orcid.org/0000-0001-5867-8760
Michele Wyler http://orcid.org/0000-0003-1097-5322
Frank LH Menke http://orcid.org/0000-0003-2490-4824
Cyril Zipfel http://orcid.org/0000-0003-4935-8583

## Decision letter and Author response
Decision letter https://doi.org/10.7554/eLife.74687.sa1
Author response https://doi.org/10.7554/eLife.74687.sa2

# Additional files

## Supplementary files
• Supplementary file 1. Transcripts predicted to encode proteins <150 amino acids up-regulated following 1 µM flg22 treatment for 90 min (*Bjornson et al., 2021*).

• Supplementary file 2. Spectral counts of peptides identified through affinity purification of the BAK1 complex.

• Supplementary file 3. Differential gene expression induced by 30 min CTNIP4$^{48-70}$ treatment.

• Supplementary file 4. Gene ontology enrichment following 30 min CTNIP4$^{48-70}$ treatment.

• Supplementary file 5. Primers used in this study.

• Supplementary file 6. Synthetic peptides used in this study.

• Supplementary file 7. Species included in CTNIP and RK search.

• Supplementary file 8. Initial CTNIP candidates used for the search.

• Supplementary file 9. Identified CTNIPs relaxed.

• Supplementary file 10. Identified CTNIPs confident.

• Supplementary file 11. Initial RK candidates used for the search.

• Supplementary file 12. Alignment of RKs identified.

• Supplementary file 13. Receptor phylogeny.

• Supplementary file 14. Full-length alignment of HAE/HSL/CEPR/RLK7/IKU2 clade.

• Supplementary file 15. Full-length HAE/HSL/CEPR/RLK7/IKU2 clade phylogeny.

• Supplementary file 16. Leucine-rich repeat (LRR) domain alignment of HAE/HSL/CEPR/RLK7/IKU2 clade.

• Supplementary file 17. Leucine-rich repeat (LRR) domain HAE/HSL/CEPR/RLK7/IKU2 clade phylogeny.

• Supplementary file 18. Kinase domain alignment of HAE/HSL/CEPR/RLK7/IKU2 clade.

- Supplementary file 19. Kinase domain HAE/HSL/CEPR/RLK7/IKU2 clade phylogeny.
- Supplementary file 20. Full-length HAESA-LIKE 3 (HSL3) alignment.
- Transparent reporting form

### Data availability

The mass spectrometry proteomics data have been deposited to the ProteomeXchange Consortium via the PRIDE partner repository with the dataset identifier PXD029264 and 10.6019/PXD029264. The RNA-seq datasets generated and analysed in the current study have been deposited in the ArrayExpress database at EMBL-EBI (www.ebi.ac.uk/arrayexpress) under accession number E-MTAB-11093.

The following datasets were generated:

| Author(s) | Year | Dataset title | Dataset URL | Database and Identifier |
|---|---|---|---|---|
| Rhodes J, Derbyshire P, Menke F | 2022 | Perception of a conserved family of plant signaling peptides by the receptor kinase HSL3 | http://www.ebi.ac.uk/pride/archive/projects/PXD029264 | PRIDE, PXD029264 |
| Rhodes J, Bjornson M | 2022 | RNAseq of Arabidopsis seedlings WT or hsl3-1, in response to a 30-min treatment of 100 nM CTNIP4 relative to mock | https://www.ebi.ac.uk/arrayexpress/experiments/E-MTAB-11093 | ArrayExpress, E-MTAB-11093 |

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
