## [Editor Report]

Beginning with transcriptome data, Rhodes et al., identify a new family of peptides with signalling function called CTNIP in the model plant *Arabidopsis thaliana*. They use an elegant biochemical capture approach to pinpoint an LRR receptor kinase called HSL3 as the only receptor for these peptides. They provide convincing genetic and biochemical evidence that HSL3 binds CTNIP and that CTNIP perception triggers HSL3-dependent cytoplasmic calcium influx, ROS production, and transcriptional changes. Furthermore, they provide initial evidence that the CTNIP-HSL3 module may participate in regulating root growth.

---

## [Decision Letter]

**Decision letter after peer review:**

Thank you for submitting your article "Perception of a conserved family of plant signalling peptides by the receptor kinase HSL3" for consideration by *eLife*. Your article has been reviewed by 3 peer reviewers, including Caroline Gutjahr as Reviewing Editor and Reviewer #1, and the evaluation has been overseen by Jürgen Kleine-Vehn as the Senior Editor.

Essential revisions:

I am appending all comments by the reviewers to help you improve your manuscript. We consider the following changes essential to make the manuscript acceptable for *eLife*:

1) Please describe the bioinformatic characterization of the CTNIP peptides in more detail (rev. #2).

2) Please describe the criteria used for design of the CTNIP4 peptides (rev. #2).

3) Include the negative control published by Hohmann et al., 2020 (rev. #2).

4) Analyze whether the CTNIP-HSL3 module plays a role in disease resistance (rev. #3).

5) Please provide a phylogenetic tree based on the ectodomain of the kinases (rev. #3).

6) Please discuss the critically that there is no evidence yet that CTNIPs are actually secreted from plant cells (rev. #3).

7) After discussion the reviewers do not demand experimental evidence that monocot CTNIP-HSL3 act as a functional module (rev. #3) but we suggest to tone down the conclusion that they are functional orthologs of Arabidopsis of the CTNIP-HSL3 module (see related comment of rev. #3).

*Reviewer #1 (Recommendations for the authors):*

I have only one suggestion regarding the last sentence of the abstract. The sentence appears a bit banal as of course a physiological role of a molecular player can only be understood, once the existence of this player is known. I suggest to think about a slightly more engaging/visionary ending of the abstract.

*Reviewer #2 (Recommendations for the authors):*

(MAJOR lines 54-58 and corresponding method section): The bioinformatic identification of AtCTNIPs is not well described in the manuscript. I could not find CTNIP-labeled genes in the supplementary material of Bjoernson et al., 2021, making it difficult to appreciate to what extend these genes were over-expressed in response to elicitor treatment, or how they were identified, grouped and aligned. I would suggest to include the AtG numbers in the main text or method section (not only in Figure 3 —figure supplement 4, CTNIP5 appears missing from this figure), together with the expression data and a more complete method section describing how these targets were selected in the revised manuscript.

(MAJOR line 64-66) Based on what criterion were the CTNIP4 fragments designed? Would it be possible that the invariant QR motif at positions 46-47 could be part of the mature peptide? In any case, it should be clearly stated in the results and conclusion section of the manuscript that the sequence of mature CTNIP4 peptide in vivo has not been determined and that it may substantially differ from the synthetic peptides used in this study. In addition, CTNIPs may be post-translationally modified (for example via hydroxyprolination of the central invariant proline residue at position 62), which may also modulate the bioactivity of the peptide. In this regard it would be helpful if the amino acid start and end positions for each peptide are clearly stated in each figure panel and in the text. To be clear I am not asking to define the mature peptide in vivo, but simply to mention this limitation of the present study.

(MAJOR Figure 3f, BIR3ecto-HSL3cyto): The authors may consider using the published negative control (BIR3ecto F146A, R170A; Hohmann et al., Plant Cell, 2020) to demonstrate signaling specificity of their chimeric receptor.

*Reviewer #3 (Recommendations for the authors):*

Just some suggestions and comments:

Line 33 'many hundreds of peptides are predicted': Good to have references for this statement.

Line 139 'similar to other LRR-RK subfamily XI signalling modules': Discussion here is too general and vague. Better to be more specific.

Line 145 'early divergent monocots': May I ask what is or what is the definition of 'early divergent monocots' here? It is necessarily to pay attention when we use of the term 'early divergent'. Same for the term 'ancient'. The article by Stuart F. McDaniel (https://doi.org/10.1111/nph.17241) can be a good read.

Line 153 'this absence is correlated with an expansion of HSL3 paralogs within these genomes': Since expansion of HSL3 in some angiosperm species is observed as well, is it appropriate to make this conclusion?

Line 155 'This is supported by the divergence between eudicot and monocot CTNIPs.': In many cases, homologs in dicots and monocots are clustered separately, as the authors also observed for several families of LRR-RLKs. And, thus, this statement might not be correct. If we draw phylogenetic trees of other known plant peptide hormones, how do they look like?

Figure 1a: Good to provide expression data of CTNIP5 as well.

Figure 2 legend title: Typo for HSL3.

Figure 3: It was rather difficult follow the order, and thus may re-organize the figure for easier reading.

Figure 4b: Typo for A. comosus.

[Editors’ note: further revisions were suggested prior to acceptance, as described below.]

Thank you for resubmitting your work entitled "Perception of a conserved family of plant signalling peptides by the receptor kinase HSL3" for further consideration by *eLife*. Your revised article has been evaluated by Jürgen Kleine-Vehn (Senior Editor) and a Reviewing Editor.

The manuscript has been improved but there are some remaining experimental issues that need to be addressed, as outlined below:

After intense discussion with the reviewers, we must insist on the control line for the BIR chimera even if this takes some time. It is important to allow concluding that the effect of the BIR chimera results from a SERK-dependent and constitutive activation of the HSL3 kinase domain, and not from other effects, such as competition of the BIR-HSL3 chimera with other LRR-RLKs that may contribute to signaling complexes controlling root development.

The trees in Figure 4 —figure supplement 1 should include all kinases shown in Figure 4a. Furthermore, you are invited to discuss the trees shown in Figure 4 —figure supplement 1 in the manuscript text.

---

## [Author Response]

Essential revisions:I am appending all comments by the reviewers to help you improve your manuscript. We consider the following changes essential to make the manuscript acceptable for eLife:1) Please describe the bioinformatic characterization of the CTNIP peptides in more detail (rev. #2).

We have now included additional information within the methods section as well as including Supplementary file 1 that shows the 99 most upregulated transcripts with gene models encoding proteins <150 amino acids following 90 min treatment with 1 μM flg22 (Bjornson *et al.,* 2021).

2) Please describe the criteria used for design of the CTNIP4 peptides (rev. #2).

We have elaborated upon this in the methods section, and thank the reviewers for highlighting this point.

3) Include the negative control published by Hohmann et al., 2020 (rev. #2).

Whilst we agree with Reviewer 2 that the BIR3_ecto_ point mutations would strengthen the argument for SERK-dependency, this would require considerable time to generate additional transgenic lines. These point mutations block the well-characterised BIR3_ecto_-SERK_ecto_ interaction (Hohmann et al., 2020). We already present considerable genetic and biochemical evidence that HSL3 functions through CTNIP-induced SERK recruitment. It would thus be surprising, but not impossible, that the phenotypes of the pHSL3::BIR3ecto-HSL3cyto-FLAG line were independent of this interaction. Accordingly, we have nevertheless discussed this within the text.

4) Analyze whether the CTNIP-HSL3 module plays a role in disease resistance (rev. #3).

We agree that this is an interesting point raised by Reviewer 3. In the current manuscript, we are reporting the initial identification of the HSL3-CTNIP signalling module and its phylogeny. Further work is now required to elucidate its physiological role(s). Under our conditions, we were unable to observe any phenotype upon spray-infection with *P. syringae* pv tomato DC3000 *ΔAvrPto/ΔAvrPto*, and have now included this data for information (Figure3—figure supplement 5). It remains to be established whether the HSL3-CTNIP signalling module contributes to resistance under different conditions or to different pathogens, or plays a role in the regulation of plant growth upon microbial perception. These are also now discussed in the text.

5) Please provide a phylogenetic tree based on the ectodomain of the kinases (rev. #3).

We have now generated phylogenies of HSL3 full-length, ectodomain and kinase domain (Figure4—figure supplement 1). We have also included the alignments and Newick files as an additional resource for readers (Supplementary file 16-21).

6) Please discuss the critically that there is no evidence yet that CTNIPs are actually secreted from plant cells (rev. #3).

We agree this is an important point and have now discussed it accordingly within the text.

7) After discussion the reviewers do not demand experimental evidence that monocot CTNIP-HSL3 act as a functional module (rev. #3) but we suggest to tone down the conclusion that they are functional orthologs of Arabidopsis of the CTNIP-HSL3 module (see related comment of rev. #3).

We have adjusted this accordingly within the text.

Reviewer #1 (Recommendations for the authors):I have only one suggestion regarding the last sentence of the abstract. The sentence appears a bit banal as of course a physiological role of a molecular player can only be understood, once the existence of this player is known. I suggest to think about a slightly more engaging/visionary ending of the abstract.

Thank you – we have adjusted the abstract accordingly.

Reviewer #2 (Recommendations for the authors):(MAJOR lines 54-58 and corresponding method section): The bioinformatic identification of AtCTNIPs is not well described in the manuscript. I could not find CTNIP-labeled genes in the supplementary material of Bjoernson et al., 2021, making it difficult to appreciate to what extend these genes were over-expressed in response to elicitor treatment, or how they were identified, grouped and aligned. I would suggest to include the AtG numbers in the main text or method section (not only in Figure 3 —figure supplement 4, CTNIP5 appears missing from this figure), together with the expression data and a more complete method section describing how these targets were selected in the revised manuscript.

We agree this is important information to include within the manuscript. We have added additional details of how the CTNIP peptide were identified both within the text and in Supplementary file 1. We have also now included the *Arabidopsis thaliana* CTNIP gene identifiers within the text. *CTNIP5* was not targeted in the *ctnip1-4* CRISPR mutant (Figure 3 —figure supplement 4).

(MAJOR line 64-66) Based on what criterion were the CTNIP4 fragments designed? Would it be possible that the invariant QR motif at positions 46-47 could be part of the mature peptide? In any case, it should be clearly stated in the results and conclusion section of the manuscript that the sequence of mature CTNIP4 peptide in vivo has not been determined and that it may substantially differ from the synthetic peptides used in this study. In addition, CTNIPs may be post-translationally modified (for example via hydroxyprolination of the central invariant proline residue at position 62), which may also modulate the bioactivity of the peptide. In this regard it would be helpful if the amino acid start and end positions for each peptide are clearly stated in each figure panel and in the text. To be clear I am not asking to define the mature peptide in vivo, but simply to mention this limitation of the present study.

We agree this is an important consideration. Identifying the sequence of the mature peptide is something we are currently pursuing, but will still require significant work, which we also like to couple with the identification of the potential proteases involved in PROCTNIP cleavage. Potentially the peptide produced in planta is longer/shorter or contains PTM, which may affect its bioactivity. Nevertheless, the data presented already show clearly that the synthetic peptides used are biologically-active and induce HSL3-dependent responses. We have discussed this further within the text.

(MAJOR Figure 3f, BIR3ecto-HSL3cyto): The authors may consider using the published negative control (BIR3ecto F146A, R170A; Hohmann et al., Plant Cell, 2020) to demonstrate signaling specificity of their chimeric receptor.

Whilst we agree with Reviewer 2 that the BIR3_ecto_ point mutations would strengthen the argument for SERK-dependency, this would require considerable time to generate additional transgenic lines. These point mutations block the well-characterised BIR3_ecto_-SERK_ecto_ interaction (Hohmann et al., 2020). We already present considerable genetic and biochemical evidence that HSL3 functions through CTNIP-induced SERK recruitment. It would thus be surprising, but not impossible, that the phenotypes of the pHSL3::BIR3ecto-HSL3cyto-FLAG line were independent of this interaction. Accordingly, we have nevertheless discussed this within the text.

Reviewer #3 (Recommendations for the authors):Just some suggestions and comments:Line 33 'many hundreds of peptides are predicted': Good to have references for this statement.

Additional references have been added.

Line 139 'similar to other LRR-RK subfamily XI signalling modules': Discussion here is too general and vague. Better to be more specific.

The phrasing has been adjusted. As many members of LRR-RK XI and their ligands have been implicated in root development, it is challenging to go into detail by citing all primary research articles. Instead, we are citing several relevant reviews.

Line 145 'early divergent monocots': May I ask what is or what is the definition of 'early divergent monocots' here? It is necessarily to pay attention when we use of the term 'early divergent'. Same for the term 'ancient'. The article by Stuart F. McDaniel (https://doi.org/10.1111/nph.17241) can be a good read.

Thank you for the suggestion. We have adjusted the text accordingly and removed reference to ‘ancient’.

Line 153 'this absence is correlated with an expansion of HSL3 paralogs within these genomes': Since expansion of HSL3 in some angiosperm species is observed as well, is it appropriate to make this conclusion?

Thank you for highlighting this. We have adjusted the text accordingly.

Line 155 'This is supported by the divergence between eudicot and monocot CTNIPs.': In many cases, homologs in dicots and monocots are clustered separately, as the authors also observed for several families of LRR-RLKs. And, thus, this statement might not be correct. If we draw phylogenetic trees of other known plant peptide hormones, how do they look like?

Thank you for the suggestion. We have adjusted the text accordingly.

Figure 1a: Good to provide expression data of CTNIP5 as well.

This data has now been included in Figure1—figure supplement 1.

Figure 2 legend title: Typo for HSL3.

Corrected

Figure 3: It was rather difficult follow the order, and thus may re-organize the figure for easier reading.

We have for the moment kept the previous figure organization as this was not flagged by other reviewers or the editors.

Figure 4b: Typo for A. comosus.

Corrected

[Editors’ note: further revisions were suggested prior to acceptance, as described below.]

The manuscript has been improved but there are some remaining experimental issues that need to be addressed, as outlined below:After intense discussion with the reviewers, we must insist on the control line for the BIR chimera even if this takes some time. It is important to allow concluding that the effect of the BIR chimera results from a SERK-dependent and constitutive activation of the HSL3 kinase domain, and not from other effects, such as competition of the BIR-HSL3 chimera with other LRR-RLKs that may contribute to signaling complexes controlling root development.

We have now generated the pHSL3::BIR3F146A,R170A ecto-HSL3cyto construct requested by the reviewers as a control and have transformed this into Col-0. In order to test for the phenotype in the T1 generation, we also generated pHSL3::LTI6B construct in the same backbone as a negative control. In the T1 generation, we could only observe the phenotype in the BIR3ecto-HSL3cyto construct, thus addressing the reviewers’ concerns.

The trees in Figure 4 —figure supplement 1 should include all kinases shown in Figure 4a. Furthermore, you are invited to discuss the trees shown in Figure 4 —figure supplement 1 in the manuscript text.

We have now repeated the analysis with all members of the HAE/HSL/CEPR/RLK7/IKU2 clade. The data is included in Figure 4—figure supplement 1 and mentioned within the text. We have also included the alignments and phylogenies within the Supplementary files should readers wish to look into theses in more detail. This aspect is also studied in some detail in a recent publication [Furumizi et al., (2021); 10.1093/plcell/koab173], which is referenced.